# Design and Synthesis of Bio-Inspired Polyurethane Films with High Performance

**DOI:** 10.3390/polym12112727

**Published:** 2020-11-17

**Authors:** Eva Marina Briz-López, Rodrigo Navarro, Héctor Martínez-Hernández, Lucía Téllez-Jurado, Ángel Marcos-Fernández

**Affiliations:** 1Instituto Politécnico Nacional-ESIQIE, Dpto. Ing. En Metalurgia y Materiales, UPALM-Zacatenco, 07738 Mexico City, Mexico; evamarinabriz@hotmail.com (E.M.B.-L.); heftor_mh@live.com.mx (H.M.-H.); ltellezj@ipn.mx (L.T.-J.); 2Institute of Polymer Science and Technology (ICTP-CSIC), Juan de la Cierva 3, 28006 Madrid, Spain; amarcos@ictp.csic.es; 3Interdisciplinary Platform for “Sustainable Plastics towards a Circular Economy” (SUSPLAST-CSIC), 28006 Madrid, Spain

**Keywords:** catechol moiety, high performance polyurethanes, hampering effect, hard and soft segment, high molecular weight

## Abstract

In the present work, the synthesis of segmented polyurethanes functionalized with catechol moieties within the hard or the soft segment is presented. For this purpose, a synthetic route of a new catechol diol was designed. The direct insertion of this catechol-free derivative into the rigid phase led to segmented polyurethanes with low performance (σ_max_ ≈ 4.5 MPa). Nevertheless, when the derivative was formally located within the soft segment, the mechanical properties of the corresponding functionalized polyurethane improved considerably (σ_max_ ≈ 16.3 MPa), owing to a significant increase in the degree of polymerization. It is proposed that this difference in reactivity could probably be attributed to a hampering effect of this catecholic ring during the polyaddition reaction. To corroborate this hypothesis, a protection of the aromatic ring was carried out, blocking the hampering effect and avoiding secondary reactions. The polyurethane bearing the protected catechol showed the highest molecular weight and the highest stress at break described to date (σ_max_ ≈ 66.1 MPa) for these kind of catechol-functionalized polyurethanes. Therefore, this new approach allows for the obtention of high-performance polyurethane films and can be applied in different sectors, benefiting from the molecular adhesion introduced by the catechol ring.

## 1. Introduction

Surface films determine the interaction of solid-state materials with their environment. Properties which can be tailored through the application of appropriate coatings are friction, adhesion, biocompatibility, and wetting, to name few. In fact, in the biomedical field, the importance of adherence is vital in certain applications. For instance, tissue adhesives must effectively function in an aqueous environment to be able to establish strong interactions with wet biological surfaces. Furthermore, an understanding of the mechanism that governs wet surface adhesion processes can assist in the development of new adhesives for its use in biological environments.

Mussels, sandcastle worms, and many other species can strongly adhere to any surface, forming interfacial bonds to various substrates for survival. This strong adhesion is primarily due to the presence of a catechol-containing amino acid called L-3,4-dihydroxyphenylalanine (L-DOPA), formed by hydroxylation of tyrosine in the structure of secreted mussel adhesive proteins (MAPs) [1,2,3,4]. In fact, Deming et al. identified that the catechol moiety in DOPA plays a key role in the universal wet adhesion property due to its chemical versatility and diversity of affinity [5].

Initial efforts to introduce these aromatic rings into polymer chains made use of radical polymerization. However, it has been established that catechol building blocks are able to scavenge free radicals by hydrogen atom transfer [6,7]. Indeed, certain compounds bearing free catechol moieties were used as polymerization inhibitors [8]. The use of monomers bearing free catechol units may also lead to a compositional drift [9,10]. This is an unavoidable feature for free radical polymerization reactions. When using this type of functional monomer, understanding the compositional drift could be vital, as the structure of the resulting copolymers may change throughout the course of the reaction, potentially leading to unintended copolymer properties and behavior [11]. To overcome this issue, the protection of the catecholic moiety has been proposed.

The influence of catecholic moieties on polyaddition or step-growth polymerizations has, to our knowledge, not been fully addressed. Although there are examples of addition polymers with catechol units in their structure, the mechanical properties of these materials were only good when crosslinking processes were used, discarding their reprocessing or recycling. For linear counterpart systems, the observed mechanical properties were very limited, dismissing their potential application in industry.

Within the breadth of addition polymers available in the industry, polyurethanes (PUs) acquire special relevance. PUs form a family of very attractive materials, being present in a wide variety of applications, including high-performance polymers. The biomedical field is one of the sectors where PUs have a marked presence in different devices, owing to the increased demand for high-performance biocompatible polymer films in their raw materials portfolio. The answer to this demand is motivated by the high versatility of these materials, capable of adjusting their structure and degradation to achieve excellent mechanical properties and biocompatibility. Indeed, these polymer systems combine excellent and attractive physicochemical properties at a competitive price [12,13]. In their micro-structure, the polymer chains are constituted by two types of partially incompatible phases. The hard segments, formed by the reaction of isocyanates and chain extenders, self-assemble via inter/intramolecular interactions (mainly hydrogen bonding) within hard domains, which are embedded within a continuous matrix of flexible soft segments built by long flexible chain diols, such as polycaprolactone (PCL).

In the present work, we innovatively introduce new catechol building blocks into the hard and soft domains of segmented PU polymers by partially replacing the classical components of PU formulations, for the purpose of obtaining PUs with improved stiffness and toughness. New strategies are presented for the insertion of catechol moieties within the polymer structure, in order to assess its influence on the degree of polymerization and, hence, the chemical and mechanical properties. Indeed, depending on the inserted form of the catechol ring (free or protected), its loading and the phase where the catechol ring is inserted and the performance of the final PU varied considerably.

## 2. Materials and Methods

### 2.1. Materials

DHBA (3,4-Dihydroxybenzoic acid), 1,1′-carbonyldiimidazole (CDI), thionyl chloride (SOCl_2_), triethyl orthoformate (TEOF), resin Amberlyst 15, polycaprolactone diol with molecular weight of 2037 (PCL-2000), 1,4-butanediol (BD), ε-caprolactone (ε-CL), tin(II) 2-ethylhexanoate (Sn(Oct)_2_**)**, N,N-dimethyl-acetamide (DMAc), and 1,6-diisocyanatohexane (HDI) were purchased from Sigma-Aldrich (Madrid, Spain). The diisocyanate product was freshly distilled at low pressure before use, and the rest of the products were used as received. AMPD (2-Amino-2-methyl-1,3-propanediol) and N,N-diethanolamine (DEA) were purchased from Merck (Madrid, Spain) and used as received.

### 2.2. Methods and Instruments

The spectra of ^1^H-NMR and ^13^C-NMR were recorded on the Bruker spectrometers Oxford 300 and Oxford 400 (Billerica, MA, USA) at room temperature (RT), using deuterated dimethyl sulfoxide (DMSO) as a solvent. Spectra were referenced with respect to the residual solvent signals at 2.50 ppm for proton spectra and 39.51 ppm for carbon spectra. Fourier transform infrared (FTIR) spectra were recorded using the PerkinElmer spectrometer model Spectrum One (Perkin-Elmer, Waltham, MA, USA) coupled with an attenuated total reflection (ATR) accessory. Sixteen scans were averaged from 4000 to 450 cm^−1^.

The molecular weights and polydispersity index (PDI) of PUs were determined by size exclusion chromatography (SEC) using a PerkinElmer apparatus (Waters Division Millipore, Madrid, Spain) equipped with an isocratic pump LC-250 and a refractive index detector series 200. A set of Styragel HR3 and HR5 Waters columns conditioned at 25 °C was used to elute 3 mg mL^−1^ samples. The mobile phase was N,N-dimethylformamide(DMF) with 0.1% of LiBr at a flow of 0.7 mL min^−1^. Polystyrene standards were used for calibration.

Thermogravimetric analysis (TGA) was performed in a Q500 calorimeter (TA instruments). The samples were analyzed in a range of temperatures from 25 to 600 °C at a heating rate of 10 °C min^−1^, under nitrogen flow. The thermal transitions were analyzed by Differential scanning calorimetry (DSC) on a Mettler Toledo DSC-822 calorimeter (Schwerzenbach, Switzerland). Samples were heated using a scanning rate of 10 °C min^−1^ in a range of temperatures from −100 to 100 °C under nitrogen purge.

Tensile properties were evaluated at room temperature using an MTS Synergie 200 testing machine, equipped with a 100 N load cell. The test speed of the machine was 5.0 mm min^−1^. The films were cut with a type 4 dumbbell die according to ISO 37. At least six samples were analyzed for each synthesized film. Results are reported in MPa.

### 2.3. Synthesis of N-(1,3-Dihydroxy-2-Methylpropan-2-yl)-3,4-Dihydroxy-Benzamide (Cat-Fun)

DHBA (10.86 mmol) and CDI (9.87 mmol) were dissolved in 100 mL acetonitrile:tetrahydrofuran (ACN:THF) (50:50) and the reaction mixture was stirred at 40 °C for 72 h under nitrogen atmosphere and protected from light to avoid oxidation of the catechol. AMPD (9.22 mmol) was dissolved in a minimum amount of ACN:THF (50:50), and then, this solution was added dropwise to the activated acid solution for 48 h. After completing the reaction, the precipitates were filtered and washed with ACN and then with THF. The solid was dried under vacuum until all solvent was completely removed.

### 2.4. Protection and Functionalization of 3,4-Dihydroxybenzoic Acid (Cat-Prot)

To a suspension of 3,4-dihydroxybenzoic acid (10.0 mmol) in 40 mL of dry toluene, TEOF (70 mmol) and Amberlyst 15 (40 mg) were added. The resulting mixture was heated at reflux for 96 h. Then, the Amberlyst was filtered off and the solvent was evaporated under reduced pressure. The resulting white solid was recrystallized from ethyl acetate. After filtration, the solid was dried under vacuum until all solvent was removed. The protected acid was then transformed into Cat-Prot following the methodology described for Cat-Fun.

### 2.5. Synthesis of Functionalized Polycaprolactone-Diol Bearing Catechol Motifs (PCL-Cat)

The preparation of this polyester diol was carried out following the standard procedure for ring opening of ε-caprolactone. In a 50-mL round-bottom flask, 2.5 mmol of Cat-Fun, 40.4 mmol of ε-CL, and three drops of SnOct_2_ were added. Subsequently, the reaction mixture was heated at 120 °C for 24 h. At the end of the reaction time and when the reaction flask was still at 120 °C, a vacuum pump was connected to remove residual monomers. Finally, the product (PCL-Cat) was obtained as a white solid and the molecular weight was determined by ^1^H-NMR. The molecular weight of this polyester was determined following the protocol described elsewhere [14]. The molecular weight determined by NMR was 1547 g mol^−1^ and the weight fraction of catechol within this polymer was 15.6%.

### 2.6. Synthesis of Polyurethanes Bearing Catechol Moieties within the Hard Segment

PUs were synthetized using DMAc as an anhydrous solvent and using 5 mol% excess of diisocyanate. As an example, the synthesis of a PU with 30% hard segments, defined as [(weight isocyanate + weight chain extender) * 100/total weight polymer], is described. In a 25-mL round-bottom flask, 1.750 g of PCL-2000 (0.859 mmol), 0.413 g of HDI (2.454 mmol), and 0.356 g of Cat-Fun (1.479 mmol) were dissolved in 3 mL of anhydrous DMAc. Subsequently, two drops of catalyst (SnOct_2_) were added. The reaction mixture was heated at 80 °C for 3 h and then it was kept at RT for 24 h. Finally, the solution was casted onto a levelled hot-plate and the solvent was slowly evaporated at 60 °C. Once the solvent had been removed, a homogeneous thick polymer film was obtained.

### 2.7. Synthesis of Polyurethane Functionalized with Catechol within the Soft-Segment

For the synthesis of this PU, functionalized PCL-Cat and pristine PCL-2000 were used at an appropriate ratio to reach a final weight fraction of catechol of 8%. In a 25-mL round-bottom flask, 0.948 g of PCL-2000 (0.465 mmol), 1.000 g of PCL-Cat (0.646 mmol), 0.104 g of BD (1.160 mmol), and 0.401 g of HDI (2.385 mmol) were added. This mixture was dissolved in 3 mL of anhydrous DMAc. Subsequently, two drops of the catalyst were added and the final reaction mixture was heated at 80 °C for 3 h. Then the polymer solution was stirred for 24 h at RT. Finally, a polymer film was obtained by casting onto a levelled hot-plate at 60 °C.

### 2.8. Preparation of Polyurethane Bearing Protected Catechol (Cat-Prot)

For the synthesis of this PU, a fraction of Cat-Prot was partially replaced by 1,4-butanediol to reach an overall content of catechol of 8% by weight. The same experimental conditions described in previous sections were used for the synthesis of this PU.

## 3. Results and Discussion

### 3.1. Synthesis of Functionalized Catechol

The insertion of catechol units within a polymer matrix is an excellent topic that has emerged with much interest in recent years because it leads to polymer materials with excellent adhesion properties. One of the simplest ways to include these catechol moieties into the PU matrix is through functionalization with aliphatic hydroxyl groups. In this sense, Duan et al. [15] performed a functionalization of the pristine catechol ring by a Mannich reaction. Our attempts, following their conditions, were unsuccessful and the yields obtained were very low. Furthermore, despite the simplicity of the starting products, two different isomers (ortho and para) were formed in the reaction, which must be separated before carrying out the polymerization reaction. As the necessary separation of these isomers requires tedious processes, this procedure was discarded for the synthesis of our target catechol.

As an alternative strategy, protocatechuic acid (DHBA) was used as a starting product to perform a suitable functionalization of the catechol. This derivative of catechol has a versatile carboxylic acid and has shown high biocompatibility, being the main polyphenol metabolite found in green tea [16,17]. Starting from this attractive acid, a functionalization focused on this carboxylic acid group was designed. This synthesis has been previously reported for the functionalization of other molecules similar to protocatechuic, such as caffeic acid [18,19]. The proposed synthesis consisted of two stages—firstly, the activation of the acid group by thionyl chloride, and secondly, the insertion of the aliphatic OH groups by the amine (AMPD) under mild conditions. The formation of the activated acid chloride (-COCl) intermediate was successfully carried out following the conditions described by Bellare et al. [20]. Nevertheless, this activated intermediate does not react selectively in the second step, leading to the formation of several products, probably owing to the high reactivity of the -COCl group towards both groups (amine and alcohol). Owing to the difficulty in purifying the reaction mass, we decided to use a more selective strategy for carboxylic acids towards amines.

In this sense, an in-situ protocol has been proposed based on the chemoselectivity of CDI towards amines. It is well-known that acyl-imidazole intermediate (**2**) (Scheme 1) reacts selectively with amines (AMPD) under mild conditions. The synthetic route is depicted in Scheme 1. The selection of this activating agent is also based on its higher activation performance in terms of scalability and greenness [21]. During the coupling reaction, only CO_2_ and imidazole are released, which can easily be removed. Indeed, CO_2_ release is the driving force for the reaction to be thermodynamically favored. Additionally, CDI presents less environmental impacts than other common alternatives, such as carbodiimides or thionyl chloride [22,23].

Once the acyl-imidazole intermediate (**2**) had formed, the amine was added directly without isolating the activated intermediate. The coupling reaction took place under the same mild conditions and, finally, the product slowly precipitated with the progress of the reaction. This facilitates the scalability of the process since the functionalized catechol (Cat-Fun) was recovered by a simple filtration and no tedious purification steps were required. Furthermore, the NMR and FTIR spectroscopic data indicated that the purity of the product was high enough to carry out the polymerization reactions.

In Figure 1, the ^1^H-NMR spectrum of functionalized catechol after filtration is shown. The purity of the obtained solid was very high and no further purification steps were required. For this reason, the product could be used directly for the synthesis of PUs. Avoiding the use of purification steps is always a key factor for the introduction of new molecules in an industrial sector [24]. The positions and multiplicities of the NMR signals correspond to those predicted—that is, above 6.5 ppm are the signals corresponding to the functionalized catechol ring and below 3.5 ppm are the signals of the reacted amine (AMPD). Furthermore, the ratio of areas between both signal groups (1:1) corresponds to the expected one, which confirms the molecular structure of functionalized catechol (Cat-Fun). Nevertheless, it is noteworthy that the hydrogen atoms of the OH groups do not show up as a sharp peak but as a diffuse peak between 4.0 and 5.5 ppm. To complete and demonstrate that the functionalization has been performed successfully, the ^13^C-NMR and the FTIR spectra were additionally performed (Appendix A).

To confirm that the isolated product was an amide, an ATR-FTIR was carried out (Appendix A). Amide bonds are easily detectable by infrared because they have several characteristic bands in very well defined regions [25]. First, the bands around 3500 and 3100 cm^−1^ correspond to the overtone of the band amide II (amide A) and the stretching vibration N-H (amide B), respectively. Additionally, in this region, the bands associated with alcohols also overlap, which implies a very pronounced widening from 3500 to 2200 cm^−1^. Of course, this widening also contains the symmetry and asymmetry of the CH_3_ and CH_2_ stretching vibrational bands. The bands amide I and amide II appear to be located at 1619 and 1524 cm^−1^, respectively, which correspond to the characteristics at the C=O tension mode of amides and in-plane N-H bending, respectively. On the other hand, the vibration bands of the catechol moiety were also detected in this spectrum. In this sense, the bands located at 1357 and 1526 cm^−1^ correspond to the ring stretching vibrations (υ_c-c_ aromatic ring) [26]; even the 1357 cm^−1^ band could have significant contributions from the amide III band. The Ph-OH bending mode was detected at 1282 cm^−1^, where it had also been detected in pristine protocatechuic acid [27]. Finally, the bands located at 1052 and 1028 cm^−1^ were associated with the C-O tension modes of primary alcohols [28].

The ^13^C-NMR of Cat-Fun is shown in Appendix A, where the peak at 169.9 ppm is characteristic of an amide carbon. The other two peaks at 148.5 and 145.5 ppm correspond to the aromatic ring carbons bearing the hydroxyl groups. Finally, the intense peak positioned at 64.5 ppm corresponds to the methylol groups (CH2OH) of the product. Its intensity is higher because both methylol groups have an equivalent magnetic environment.

### 3.2. Synthesis of Polyurethanes Bearing Catecholic Moieties within the Hard Segment

The aim of introducing these catecholic units into a polymer matrix is to mimic the proteins responsible for mussel adhesion (MAPs) on any organic or inorganic substrate [29]. Indeed, catecholic units are the key elements to provide a correct adhesion of the polymer to the substrate. In addition, polymer films must possess advanced features that allow good mechanical properties to be obtained, as well as excellent coating-forming properties. In this sense, PUs emerge as polymers with a very promising future because they are very versatile polymers and can be easily modulated. In the literature, there are few examples of PUs with catechol motifs within their microstructure [15,30,31,32]. In those works, the catechol loading content varies widely between 8% and 20% by weight; however for highly crosslinked systems, the content is triggered to 60%. In the work carried out by Panchireddy et al. [31], the insertion of these active catecholic moieties was performed by combining two polymerization methods; one was the formation of free-isocyanate PUs and the other was the oxidative self-polymerization of dopamine. It is well-known that polydopamine [33] confers a characteristic black color to its polymers, which modifies their optical properties and could impede its use as a coating in certain applications. On the other hand, the main characteristic in those works focused on linear and soluble PUs with relatively low molecular weights (<30,000 g/mol) that use poly(ethylene glycol) (PEG) as a soft segment, which lead to PUs with poor mechanical properties [34,35]. In fact, in all these works, no mechanical properties of the resulting PUs were reported. Therefore, to our knowledge, a systematic study analyzing the influence of catechol units on the composition of PUs has not been addressed. The evaluation was carried out mainly through the variation of the mechanical properties with respect to the hard segment or catechol content.

Hence, a series of PUs bearing catechol motifs were prepared, using Cat-Fun as a new chain extender. The synthesis and structure of PUs are shown in Scheme 2. The other components required for the synthesis of PUs were PCL-2000 as a soft segment and HDI as a diisocyanate, because both components are the basis of many biocompatible materials with excellent mechanical properties [36,37].

Firstly, three different PUs were prepared with increasing catechol content. In Table 1, the weight fractions and the molecular weights of the synthetized PUs are collected. As Cat-Fun played the role of chain extender, an increase in the catechol content led to a corresponding increase in the hard segment content. These synthesized PUs covered a range of hard segment content widely used in industry, which varies from 20% to 40% by weight [24].

This variation in chemical composition leads to a modulation of the chemical and mechanical properties of the final PUs. According to the data shown in Table 1, as the hard segment content increased, the molecular weight of the PUs decreased progressively. In fact, a significant change between PU-Cat HS 20 and PU-Cat HS 30 was observed, and the average molecular weight (Mn) dropped significantly to half of its value. This could indicate that the catechol-based chain extender exerted a negative influence on the polyaddition reaction. This inhibitory or retarding effect of catecholic moieties has been previously addressed for radical polymerization reactions [38]; however, works advocating a similar effect of these building blocks in polyaddition reactions have not been found.

Cat-Fun could, theoretically, behave like a tetrafunctional compound, with two phenol and two aliphatic hydroxyl groups. In principle, this should lead to chemical crosslinking during the polymerization reaction if the average functionality is greater than two. However, the calculated average functionality was two for these polyurethanes and only at full conversion would a network be formed. It was found that all synthesized PUs were perfectly soluble in standard solvents, such as DMF, DMAc, or DMSO. It is well-known that the reaction between an aromatic hydroxyl group (Ph-OH) and isocyanate groups occurs reversibly [38]—in fact, this is the *key element* of some smart self-healing polymers [39,40,41]. However, in the specific case of catechol units, their phenolic hydroxyl groups do not react with isocyanates; this fact has been previously addressed by Duan et al. [15] Therefore, the compound Cat-Fun should react exclusively by means of its alkyl-OH groups, keeping the catecholic unit unaltered. Nevertheless, it could not be discarded that phenolic groups exerted a hampering effect on the growth of the chain.

The structure of PUs bearing catecholic units was confirmed by FTIR-ATR, ^1^H-NMR, DSC, and TGA. In Figure 2, the ATR-FTIR spectra of functional PUs and Cat-Fun are shown. The band located at 1724 cm^−1^ was attributed to the stretching vibration mode of the carbonyl group C=O in the polyester (PCL, soft segment); in addition, an overlapping shoulder corresponding to the vibration of carbonyl groups from urethanes was also observed [42]. A series of bands at 1234, 1176, and 1091 cm^−1^ were found related to vibrations of the PCL-backbone segment (C-O-C). Additionally, the presence of the catecholic units was confirmed by the band located at 1526 cm^−1^, corresponding to the tension of the aromatic ring (υ_arC-C_) [43].

To determine the thermal properties of these segmented PUs, the samples were analyzed by DSC. All discussion is based on the second heating scan, after melting of the materials to erase their thermal history. Figure 4 shows the DSC measurements for the synthetized segmented PUs with catecholic building blocks within the hard segment.

In the case of PU-Cat-HS 20, only two thermal phenomena were observed, while three thermal transitions were detected for its analogues. For the three systems, firstly, a glass transition temperature (T_g_) of the soft segment (PCL) was detected at −59 °C and then a melting temperature (T_m_) of crystalline phase was found at +61 °C. Because these two thermal transitions appeared at the same temperature for the three segmented PUs, it was possible to associate these transitions with the soft segment, since the distribution and molecular weight of soft segment chains (PCL-2000) were identical in the three polymers. However, the degree of crystallinity of this soft segment decreased significantly with respect to pristine PCL-diol (Appendix A), probably owing to the reduced mobility of end-capped hydroxyl groups of the PCL-diol chain after being incorporated into the PU chain.

The PUs with a higher hard-phase content (30% and 40%) additionally showed an endothermic peak linked to the crystallization of the soft domains. The polymer morphology for these two polymers could induce a favorable crystallization environment and a lower distribution and molecular weight. This finding is correlated with the observed Gel permeation chromatography (GPC) data. The polymer chains of these lower molecular weight PUs have higher molecular mobility, thereby inducing crystallization of the soft phase during heating scan. This thermal transition occurred at different temperatures; for polymer PU-Cat-HS 30, the peak appeared at −6 °C, whilst for the polymer PU-Cat-HS 40, this temperature rose to 11 °C. This difference could be due to the hindering effect of the hard segment and the catecholic content during the crystallization of the soft segment, whereby an increase in the hard segment content (and catechol moieties) leads to a more impeded crystallization produced at higher temperatures.

The thermal decomposition of the segmented PUs is shown in Figure 5. TGA analysis showed that the polymers PU-Cat-HS 30 and PU-Cat-HS 40 started their thermal decomposition with an inflexion temperature (T_d1_) at 207 and 210 °C, respectively. Subsequently, these polymers showed a second inflexion temperature (T_d2_) at which almost the total loss of polymer occurred, leaving 2% of polymer residue. However, at around 350 °C, polyurethane PU-Cat-HS-20 showed the most significant weight loss, losing 90% weight of the degraded polymer. Previously, in this polymer, a slight decrease in weight (5.4%) at over 230 °C was observed. The inflexion temperatures of each PU are collected in Appendix A.

Although there is no obvious assignment for the degradation steps of these segmented PUs, the relative weight loss at each stage could be taken as evidence to relate the degradation stages with either one segment or the other. On this basis, we have assigned the first inflexion temperature (T_d1_) to the hard segment and the second one (T_d2_) to the soft domain, that is, the thermal degradation of the PCL block.

The mechanical properties of the PUs were measured in tension. Due to the difficulty of handling PUs with a high hard segment content (30% and 40%) during the die-cutting stage, the mechanical properties of these polymers were not measured. Indeed, this behavior was associated with the low molecular weights obtained during the polymerization stage; this fact was in agreement with the GPC data. Therefore, the interpretation of the mechanical properties of polyurethane PU-Cat-HS 20 will be addressed and discussed in detail in a later section through a comparison with other similar systems.

Until now, the results of PUs functionalized with catecholic building blocks within the hard segment have been presented. Only for PU-Cat-HS 20 was the molecular weight obtained relatively high, and for this reason, the mechanical properties for the other PUs with a higher hard segment content remained low, discarding any possible interest of the industry in this type of adherent films. To address this drawback, two different alternatives were suggested. The first proposal focused on the synthesis of segmented PUs by inserting the catecholic units within the soft segment, while the second proposal was based on the protection of these building blocks. For both initiatives, the composition of the PUs remained constant—that is, 20% of the hard segment and 8% catechol content. For this reason, the reference system will be PU-Cat-HS 20.

### 3.3. Synthesis of Polyurethanes Functionalized with Catechol Units within the Soft Segment

For the first time, the insertion of catechol building blocks into the soft segment of a PU was addressed. The dynamics and molecular mobility of these active units within this soft domain should be higher, favoring the specific interactions of catecholic motifs with their surrounding environment. To carry out this insertion, a previous stage of functionalization of the soft segment was required. This methodology was based on the ring-opening polymerization of ε-caprolactone using Cat-Fun as the initiator (Scheme 3). The ratio between monomer and initiator determined the molecular weight of the polyester (PCL-Cat), which was modulated to obtain a molecular weight similar to the reference system (PU-Cat-HS 20).

In Appendix A, the proton NMR spectrum of the polyester (PCL-Cat) carrying catechol units is shown. The peak area ratio between 4.0 and 3.6 ppm dictated the molecular weight of the polyester. Through appropriate derivatization with trifluoroacetic anhydride [14], the calculated molecular weight was 1547 g mol^−1^. Additionally, in this spectrum, the aromatic signals from the initiator were also detected, demonstrating that the ring-opening reaction had properly worked.

Subsequently, when this functionalized polyester bearing catecholic moieties was directly used in the formulation of the functionalized PU within the soft segment (PU-Cat-SS 20), the minimum catechol loading in the final PU without adding a chain extender would be higher than 8% weight, and for this reason, it was necessary to include pristine PCL-2000 diol into the formulation. Additionally, to ensure that catechol was present only in the soft segment, Cat-Fun was formally replaced by 1,4-butanediol (BD). This chain extender is one of the most widely used by the PU industry [24].

The ^1^H-NMR spectrum of polyurethane PU-Cat-SS 20 and a comparison between the ATR-FTIR spectra of functionalized PUs bearing catecholic moieties within the hard and soft domains are depicted in Appendix A. The presence of catecholic units within the soft segment was confirmed by the detection of aromatic hydrogens at around 7 ppm (in Appendix A) and the band located at 1357 cm^−1^ (in Appendix A) from the stretching vibration of aromatic ring of the protocatechuic acid derivate.

### 3.4. Synthesis of Polyurethanes with Protected Catechol Units within the Hard Segment

Catecholic units can be easily protected by a wide variety of protective groups, including acetonides [45], cyclic carbonates [46], and cyclic boronates [47]. However, the orthoformate group has special relevance because the deprotection reaction requires mild conditions in an aqueous medium [48,49]. This aspect is decisive to minimize the degradation effects of the backbone polymer and to attain an efficient deprotection of the active catechol unit. The synthesis route used for the preparation of the catechol protected with the orthoformate group (Cat-Prot) is shown in Scheme 4.

For the synthesis of Cat-Prot, firstly, the orthoformate protecting group was introduced using triethyl orthoformate as a protecting group source and Amberlyst 15 resin as an acid catalyst. A complete NMR characterization of this acid intermediate is collected in Appendix A. Once the aromatic ring was protected, the same synthetic steps described for Cat-Fun were followed. In Appendix A, the NMR spectra (^1^H and ^13^C) of the product Cat-Prot are depicted. The insertion of the protecting group was confirmed by the presence of the singlet located at 7.11 ppm (Appendix A) and the peak at 118.9 ppm (Appendix A). Both signals correspond to the characteristic CH group of the orthoformate rest. As in the case of Cat-Fun, the purity of the protected product was high enough to introduce it directly into the PU formulations.

For the preparation of PUs bearing protected catechol motifs within the hard segment, Cat-Prot was used as a chain extender and the rest of the components employed were similar to those described above. For this new PU, the hard segment content was also set to 20%, which made it possible to compare these results with the PUs described above. The NMR spectrum of PU-Cat_Prot_-HS 20 is shown in Figure 6. Although the signals were very similar to the PU bearing free catechol units, certain significant differences were noted. Firstly, the signal positioned at 7.11 ppm was maintained, and secondly, the ethoxy side group (-OCH_2_CH_3_) appeared at 3.70 and 1.02 ppm, respectively. Both sets of signals confirmed that the orthoformate protecting group was stable and compatible with the polymerization conditions.

### 3.5. Influence of Catechol Moieties on the Polyurethane Properties

The insertion of catechol units within a polymeric matrix has maintained the transparent character of the films. In Appendix A, pictures of the three catechol-functionalized polymers and the PUs lacking catechol units are shown. This reference PU was obtained by using 1,4-butanediol instead of Cat-Fun as a chain extender. Transparency is a characteristic that differs from polydopamine-based systems, which introduce a characteristic black color and could impede their use in certain applications.

The thermal transitions, the elastic moduli, and the molecular weight of synthetized PUs (PU-Cat-HS 20, PU-Cat-SS 20, and PU-Cat_Prot_-HS 20) are collected in Table 2. All PUs were perfectly soluble at RT in standard solvents, such as DMF, DMAc, and DMSO. This discards that any chemical crosslinking side reactions had taken place.

On the one hand, the formal displacement of the catechol ring from the hard segment (PU-Cat-HS 20) to the soft domain (PU-Cat-SS 20) favored the polymerization reaction, reaching two-fold higher molecular weights. In this way, the inhibitory effect induced by the catechol motif could be reduced, probably due to the increased distance between the building block and the chain ends in PCL-Cat. As a result, the degree of polymerization was considerably increased and, probably, PU domains will be effectively segregated. On the other hand, the protection of the aromatic ring (PU-Cat_prot_-HS 20) greatly favored the degree of polymerization, leading to the highest molecular weights. Despite the proximity between the catecholic aromatic ring and the chain ends (R-OH), the achieved molecular weights were four-fold higher than that of the reference system (PU-Cat-HS 20).

To shed light on these behaviors, our hypothesis is based on the tendency of catechol and its derivatives to oxidation reactions [50,51]. Indeed, in the particular case of protocatechuic acid and its derived esters, they have been described to undergo oxidation reactions with R-XH groups (being XH alcohols, amines, or thiols) [52,53]. Therefore, when the catechol derivative was located within the hard segment (Cat-Fun), the proximity of the aromatic ring to the chain ends (alkyl-OH groups) could lead to side reactions, leading to stoichiometric imbalances or hindering the growth of the polymer chain in PU-Cat-HS 20. Nevertheless, through the orthoformate group in Cat-Prot, the oxidation reaction of the catechol group was effectively inhibited, allowing the polymer chain to reach the highest molecular weights. Possibly, the Cat-Prot chain extender lacks propensity to oxidation, inhibiting side reactions associated with protocatechuic acid derivatives that would lead to stoichiometric imbalances. Despite the considerable increase in molecular weight associated with the protection of the catechol group, the polydispersity remained practically constant. This could reinforce the proposed theory about the hampering effects associated with aromatic OHs during chain growth.

The thermal and mechanical behaviors of the studied PUs are shown in Figure 7.

In Figure 7A, the main difference between these polymeric materials resided in the presence of an endothermic transition, shown by PU-Cat-SS 20. This transition was located at 0 °C and it was assigned to the crystallization of the soft segment. Additionally, regarding the amorphous phase, the T_g_ (−52 °C) of catechol-bearing PU in the soft domain was slightly higher than that of its partners. This could be mainly due to the lower molecular weight of the functionalized soft segment (PCL-Cat) compared to pristine PCL-diol. Furthermore, related to this issue, the soft domain of PU-Cat-SS 20 presented a lower crystallinity. In this case, the presence of the catechol moiety within this phase could hinder the correct crystallization of the polycaprolactone chains. On the other hand, the PU carrying free catechol (PU-Cat-HS 20) presented the highest crystallinity, probably due to the lower relative molecular weight, which leads to higher molecular mobility. Moreover, the orthoformate protecting group of PU-Cat_prot_-HS 20 could establish additional interactions with urethane groups, preventing a correct crystallization of the soft phase.

In Figure 7B, the stress–strain curves for the three prepared PUs are compared. The PU-Cat-HS 20 revealed very different mechanical behavior from its counterparts. While this material resembled a thermoplastic PU, the other two polymers responded with an elastomeric behavior. The stress modulus increases linearly with respect to deformation until it reaches its maximum. There are also significant differences related to the Young’s modulus, with polymer PU-Cat-HS 20 giving a high modulus of 159 MPa, characteristic of thermoplastic materials, while PU-Cat-SS 20 and PU-Cat_prot_-HS 20 were characterized by their low initial moduli (~9 MPa). Surprisingly, the maximum stress at break for the PU carrying the protected catechol (PU-Cat_prot_-HS 20) was 66.1 MPa, representing a new maximum for PUs functionalized with catechol units. Indeed, this new maximum was two-fold higher than results previously published for other equivalent systems. For instance, Filippidi et al. [54] indicated a value of 35.0 MPa, and Panchireddy et al. [31] reported 34.2 MPa for their catechol-based PUs. Furthermore, it should be noted that in those works, the PU systems were strongly cross-linked, so it was expected that their mechanical responses would be much superior to those of linear systems. However, for our linear system, the protection of the catechol group and the high degree of polymerization achieved have led to these attractive mechanical properties. These characteristics conferred a high added value and a differential element to this system compared to its competitors.

## 4. Conclusions

Functionalization of protocatechuic acid has been successfully performed by effective CDI-activated coupling. The synthetic route designed for the catechol derivative (Cat-Fun) showed high yield under mild conditions and also easy handling. The use of the product Cat-Fun as a chain extender led to low-performance polyurethane, limiting its loading content to 8% by weight. As the catechol content increased, the molecular weight and mechanical properties dropped sharply. Probably owing to the hampering effect of Cat-Fun during the polymerization reaction, the increase in the degree of polymerization was prevented.

The high versatility of Cat-Fun allowed for the ring-opening of ε-caprolactone, leading to a polyester carrying catechol moieties. This functionalized polyester was subsequently introduced into the formulation of a polyurethane bearing catecholic moieties within the soft segment. The insertion of this building block into the soft phase allowed for achieving very interesting mechanical properties, due to the increased degree of polymerization, probably because of decreased inhibition of the catechol ring. However, the protection of the catecholic ring with an orthoformate group implied a very significant increase in the molecular weight of the final polyurethane. The absence of side reactions caused by the catecholic ring would account for the complete suppression of inhibitory effects during the polymerization reaction and the increase in the degree of polymerization. The mechanical properties of this polyurethane were consequently triggered at 66.1 MPa (maximum stress at break), obtaining an elastomeric material with high added value and high toughness. This new maximum represents a new milestone for polyurethane films carrying catecholic units.

Through this protection approach, the possibility of increasing the content of catechol within this type of segmented polyurethanes has been opened, while maintaining excellent mechanical properties. This task is currently being carried out in our laboratory, with the aim of evaluating the influence of catechol content on the mechanical properties and the adhesion capacity of these polymer films.

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
