# Peer review of "Design and Synthesis of Bio-Inspired Polyurethane Films with High Performance"

_polymers, 2020, doi:10.3390/polym12112727_

Round 1

Reviewer 1 Report

The manuscript entitled “Design and synthesis of bio-inspired polyurethane films with high performance” has been reviewed. The results are interesting and helpful. However, the paper needs to be revised before acceptation. Detailed comments are as follows:

  1. There are several typo errors in the manuscripts. For e. g., “-1” in “min-1” should be in subscript form. “figure 1” should be “Figure 1”. Please pay attention to unnecessary capital of some words, like Tissue Adhesives, Polycaprolactone. Please recheck your manuscript carefully.
  2. The abbreviation, such as PUs should be given the full name where it first appeared. Additionally, once the abbreviation is defined, it should be used in the following text.
  3. The full names of abbreviations, such as DMSO, MTS, ACN, THF, should provide full names where they first appeared.
  4. The abbreviations, such as DHBA, CDI, should not be defined more than once.
  5. In Figs. 1, 3 and 6, axis names should be provided.
  6. In Fig. 4, Heat flow should be Heat Flow.
  7. In Fig. 4 and Fig. 7, the DSC thermograms were obtained the first heating run. Since the previous physical aging of a polymer affects the DSC results, the thermal history should be erased by annealing the sample at a certain temperature for a certain period. So, the second heating run was used to determine the DSC results (M. Wagner, Thermal Analysis in Practice: Fundamental Aspects, Hanser Publishers, Munich, Germany, 2017.).
  8. In Fig. 2, the vertical space between two spectra should be wider.
  9. In Tables 1 and 2, D should be PDI.
  10. In Table 2, standard deviations should be added to tensile test results.

Author Response

The manuscript entitled “Design and synthesis of bio-inspired polyurethane films with high performance” has been reviewed. The results are interesting and helpful. However, the paper needs to be revised before acceptation. Detailed comments are as follows:

  • There are several typo errors in the manuscripts. For e. g., “-1” in “min-1” should be in subscript form. “figure 1” should be “Figure 1”. Please pay attention to unnecessary capital of some words, like Tissue Adhesives, Polycaprolactone. Please recheck your manuscript carefully.

The authors are grateful for the changes proposed by the reviewer and the manuscript has been carefully reviewed. The proposed changes have been highlighted in yellow.

  • The abbreviation, such as PUs should be given the full name where it first appeared. Additionally, once the abbreviation is defined, it should be used in the following text.

We appreciate the comment made by the reviewer and this assessment has been addressed on page 2 (third paragraph) of the revised version of the manuscript.

  • The full names of abbreviations, such as DMSO, MTS, ACN, THF, should provide full names where they first appeared.

The changes proposed by the reviewer have been conveniently collected. All proposed changes have been highlighted in yellow.

  • The abbreviations, such as DHBA, CDI, should not be defined more than once.

We appreciate the reviewer's comment and the changes have been made conveniently.

  • In Figs. 1, 3 and 6, axis names should be provided.

The authors of the manuscript appreciate the observation made by the reviewer. However, the figures mentioned by the reviewer correspond to NMR spectra for the synthetized compounds. The representation of these spectra follows the usual format for the presentation of these figures, that is, only the x-axis is represented. This axis corresponds to the chemical shift and is abbreviated as delta (δ) and its units are ppm.

  • In Fig. 4, Heat flow should be Heat Flow.

This correction has been made in the revised version of the manuscript.

  • In Fig. 4 and Fig. 7, the DSC thermograms were obtained the first heating run. Since the previous physical aging of a polymer affects the DSC results, the thermal history should be erased by annealing the sample at a certain temperature for a certain period. So, the second heating run was used to determine the DSC results (M. Wagner, Thermal Analysis in Practice: Fundamental Aspects, Hanser Publishers, Munich, Germany, 2017.).

The authors appreciate the observation made by the reviewer, which is in line with that made by the second reviewer. All the results and interpretations made on the thermograms actually corresponded to the second scan. To improve the interpretation of the results, the following clarifying phrase has been added on page 9:

All discussion is based on second heating scan, after melting of the materials to erase their thermal history.

  • In Fig. 2, the vertical space between two spectra should be wider.

We appreciate the comment provided by the reviewer, and this has been carried out.

  • In Tables 1 and 2, D should be PDI.

The authors of the present manuscript appreciate the reviewer's comment. However, we believe that the symbol used to describe the polydispersity of polymers is not a "simple" D, the symbol used is "D-Stroke" which is the symbol accepted by the IUPAC to refer to polydispersity. In fact, in the following reference (Pure Appl. Chem., 2009, 81(2), 351-353) it can be read: “The general symbol Đ, pronounced “D-stroke”, is introduced for dispersity to avoid confusion with the conventional use of D for diffusion coefficient

We kindly invite the reviewer to check the reference: Pure Appl. Chem., 2009, 81(2), 351-353

  • In Table 2, standard deviations should be added to tensile test results.

We appreciate the comment made and the standard deviations have been conveniently included in the table.

Reviewer 2 Report

Dear Authors,

congratulation to this paper. I have only one recommendation:

Please add also in the method part, that you use the second heat up o f the DSC.

BR

The reviewer

Author Response

Dear Authors,

congratulation to this paper. I have only one recommendation:

Please add also in the method part, that you use the second heat up o f the DSC.

Los autores agradecen la amabilidad mostrada por el revisor y su acertado comentario sobre la necesidad de indicar explícitamente que los resultados de DSC se corresponden con el segundo barrido. Es por ello, que esta apreciación ha sido abordada correctamente en la versión corregida del manuscrito, incluyendo la siguiente frase en la página 9. Adicionalmente, este comentario está alineado con el comentario realizado por otro revisor de este manuscrito.

The authors appreciate the kindness shown by the reviewer and his/her wise comment on the need to explicitly indicate that the DSC results correspond to the second scan. Therefore, this assessment has been correctly addressed in the revised version of the manuscript, including the following sentence on page 9. Additionally, this comment is aligned with the comment made by another reviewer of this manuscript.

All discussion is based on second heating scan, after melting of the materials to erase their thermal history.

Round 2

Reviewer 1 Report

The manuscript entitled “Design and synthesis of bio-inspired polyurethane films with high performance” has been reviewed again. Some comments were not well considered. So, the manuscript needs to be well revised. Detailed comments are as follows:

  1. There are several typo errors in the manuscripts. “figure 1” should be “Figure 1” in Line 203. Please recheck your manuscript carefully.
  2. The abbreviation, such as PUs in Line 62 should be given the full name where it first appeared. Additionally, once the abbreviation, such as CDI is defined, it should be used in the following text.
  3. The full name of abbreviations, such as DMSO, CAN and THF, should be provided where they first appeared.
  4. In Figs. 1, 3 and 6, axis names and units should be Chemical shift (ppm).
  5. In Fig. 2, the vertical space between two spectra should be wider. The authors did not translate these spectra vertically.
  6. In Table 2, standard deviations in form of positive-negative signs should be added to tensile test results.

Author Response

The manuscript entitled “Design and synthesis of bio-inspired polyurethane films with high performance” has been reviewed again. Some comments were not well considered. So, the manuscript needs to be well revised. Detailed comments are as follows:

1. There are several typo errors in the manuscripts. “figure 1” should be “Figure 1” in Line 203. Please recheck your manuscript carefully.

We have rechecked the manuscript carefully to our best, and corrected all the typo errors we have found.

2. The abbreviation, such as PUs in Line 62 should be given the full name where it first appeared. Additionally, once the abbreviation, such as CDI is defined, it should be used in the following text.

We have used abbreviations throughout the text once it has been defined. In the case of the word polyurethane, that appears profusely on the text, once the abbreviation was defined, we have used the abbreviation in general except for Figure captions, Table headings, sections headings, conclusions and in some particular cases.

 3. The full name of abbreviations, such as DMSO, CAN and THF, should be provided where they first appeared.

The changes proposed by the reviewer have been conveniently collected. All proposed changes have been highlighted in yellow. (For DMSO, please check line 96, and for ACN and THF line 117)

4. In Figs. 1, 3 and 6, axis names and units should be Chemical shift (ppm).

The authors thank the reviewer for his assessment, and the figures have been updated following his indications.

5. In Fig. 2, the vertical space between two spectra should be wider. The authors did not translate these spectra vertically.

We appreciate the reviewer's comment and the distance between the spectra has been conveniently increased to facilitate reading and interpretation.

6. In Table 2, standard deviations in form of positive-negative signs should be added to tensile test results.

Standard deviations have been added to the results in Table 2.

Round 3

Reviewer 1 Report

The manuscript has been well revised. It can be accepted now.